# A GUIDE TO MISINFORMATION DETECTION DATASETS

## ABSTRACT

Misinformation is a complex societal issue, and mitigating solutions are difficult to create due to data deficiencies. To address this problem, we have curated the largest collection of (mis)information datasets in the literature, totaling 75. From these, we evaluated the quality of all of the 35 datasets that consist of statements or claims. We assess these datasets to identify those with solid foundations for empirical work and those with flaws that could result in misleading and non-generalizable results, such as insufficient label quality, spurious correlations, or political bias. We further provide state-of-the-art baselines on all these datasets, but show that regardless of label quality, categorical labels may no longer give an accurate evaluation of detection model performance. We discuss alternatives to mitigate this problem. Overall, this guide aims to provide a roadmap for obtaining higher quality data and conducting more effective evaluations, ultimately improving research in misinformation detection.

## 1 INTRODUCTION

Misinformation is a pressing concern for society, already causing significant negative impacts and posing even greater risks with the advent of generative AI (Torkington, 2024). Extensive research has been devoted to this problem, yet it remains unresolved. There has been considerable recent progress in methods, especially leveraging LLMs (Chen & Shu, 2023). However, to fuel further progress, we also need strong and reliable data.

Multiple studies have identified data availability, and especially data quality, as a barrier in this domain. To begin with, obtaining high quality veracity labels is challenging and time-consuming, even for experts (Zubiaga et al., 2016). Shortcuts, though, can cause severe spurious correlations (Pelrine et al., 2021; Wu & Hooi, 2022), and even with high quality labels there can be issues with ambiguity of input texts (Pelrine et al., 2023). There are many surveys of the method landscape (Shu et al., 2017; Oshikawa et al., 2018; Zhou & Zafarani, 2020; Chen & Shu, 2023), but analysis of datasets has often been limited in scale (Pelrine et al., 2021; Wu & Hooi, 2022; Pelrine et al., 2023) or depth.

In this study, we present the largest scale survey of datasets in the literature to date, curating 75 datasets along with corresponding descriptive analyses and categorizations. This is nearly 3 times as many as other dataset-focused surveys like Hamed et al. (2023), and many times more than general surveys like Ali et al. (2022); Shu et al. (2017); Oshikawa et al. (2018); Zhou & Zafarani (2020). We provide a summary of each dataset, along with key information like topic, size, modality, languages, geographic region, and time period.

Then, we focus on 35 datasets that include claims, which serve as key sources of atomic data for misinformation detection, and we evaluate their quality in depth. First, we assess label quality, examining the labeling process and comparing the advantages and limitations of expert fact-checking, crowdsourced labels, source-based evaluations and algorithmic labeling. Then, we examine three types of potential spurious correlations and bias, that could lead to predictions based on invalid, non-generalizable signals. In particular, we start by looking at keyword based correlations—which can also serve as a proxy for topic or event. Then we look at temporal correlations, and finally the political leaning of the statements. These quality evaluations, combined with our descriptive analysis, offer practical insights for selecting datasets in future misinformation research.

Once datasets are chosen, implementation and evaluation questions remain. To help address these, we first present a unified formatting and labeling schema for all 35 claims datasets. Next, we establish

state-of-the-art baselines using GPT-4 with and without web search to collect evidence. Following this analysis, we find that standard evaluation metrics like accuracy and F1, when computed simply in relation to ground truth labels, are no longer sufficient to evaluate leading generative methods for misinformation detection and could lead to invalid conclusions. We provide initial work on an alternative evaluation as a starting point for future research, and suggest such evaluation research is urgently needed in the misinformation detection domain.

In summary, we present a guide to misinformation datasets, including:

- The largest scale survey of such datasets, with over 75 identified and analyzed.
- An in depth evaluation of the quality of 35 datasets focused on claims, identifying limitations to take into account when selecting and using them.
- Practical tools for research using those datasets: a unified file and label schema, and state-of-the-art baselines.
- Analysis of evaluation suggesting simple metrics like accuracy and F1 may be obsolete in this domain, and initial work on alternatives.

We provide our code, unified data, and other outputs on GitHub and OSF.[1][2]

## 2 RELATED WORK

In recent years, the scientific community has shown a growing interest in fake news detection to mitigate the spread of false information. Within this evolving field, several surveys have emerged to offer comprehensive reviews and standardized evaluations. A pioneering effort by Shu et al. (2017) provided an early framework, defining fake news, detailing its characteristics, and summarizing detection techniques from a data mining perspective. Subsequent surveys, such as Oshikawa et al. (2018) and Zhou & Zafarani (2020), have explored alternative methodologies, focusing respectively on natural language processing (NLP) methods and interdisciplinary perspectives. However, while these surveys and others (Bondielli & Marcelloni, 2019; Gravanis et al., 2019) are valuable for providing a comprehensive overview of the state-of-art in fake news detection, they pay limited attention to existing datasets. Indeed, even if some emphasize the challenges of data collection or stress the importance of dataset quality, these surveys usually provide only superficial coverage of existing datasets, overlooking their specific content, details, and characteristics. Assessing the quality of these misinformation datasets is critical because they are often used to train and test models for misinformation detection and related tasks. A lack of quality data in this context implies that biases and erroneous conclusions could be introduced both in the development and in the validation process of these systems.

This gap has thus spurred the emergence of additional surveys dedicated to addressing these dataset-centric nuances, which can be categorized into two types. The first one focuses on categorizing existing datasets to guide the research community in their selection. For example, D'Ulizia et al. (2021) surveyed 27 datasets based on eleven characteristics (e.g., application purpose, type of disinformation, language, size, news content type, etc.) and compared these quantitatively. Another example, Sharma et al. (2019) summarized the characteristic features of 23 existing datasets, providing a clearer picture of those available to the public. However, these surveys have an important drawback; they often lack in-depth analysis. In fact, only descriptive characteristics are listed, thus neglecting key characteristics of their quality and effectiveness for future research. This is also the case for Ali et al. (2022) and Patra et al. (2022), which describe 26 and 7 datasets, respectively.

The second type of survey focuses on analyzing the quality, performance, and limitations of datasets. For instance, Abdali (2024) examines 10 datasets to identify some of these weaknesses and strengths. However, a broad approach is used to outline biases, which fails to detail the specifics of each dataset, leaving researchers uncertain about their individual quality. Another example, Hamed et al. (2023) highlight the limitations of 20 articles using publicly available datasets. While this approach provides a good overview of literature trends, a grey area remains regarding whether the errors in these 20 articles stem primarily from methodology or dataset issues. We also find the work of Pelrine et al. (2021), who evaluate the quality of six datasets, focusing on their potential spurious correlations with temporal information. Wu & Hooi (2022) expands the analysis of the spurious correlations issue to those induced by event-based collection, dataset merges, and labeling bias, using the Twitter15,

---

[1] https://anonymous.4open.science/r/misinfo-datasets-3AB5/README.md
[2] https://osf.io/5azde/?view_only=cf103519a4454286becf5699f85bd77b

Table 1: Characterizing 75 common misinformation datasets. Datasets are ordered by modality, then date, and topic.

| Dataset | Size | Modality | Topic | Geographic region | Language | Time start | Time end |
|---|---|---|---|---|---|---|---|
| AntiVax | 15,465,687 | Claims | Health | USA | English | 01/12/2021 | 31/07/2021 |
| CoAID | 301,177 | Claims | Covid-19 | – | English | 01/12/2019 | 01/09/2020 |
| Counter-covid-19-misinformation | 155,468 | Claims | Covid-19 | International | English | 21/01/2020 | 20/05/2020 |
| COVID-19-Rumor | 7,179 | Claims | Covid-19 | – | English | 01/2020 | 03/2020 |
| Covid-19-disinformation | 16,000 | Claims | Covid-19 | International | 4 languages | 01/2020 | 03/2021 |
| Covid-vaccine-misinfo-MIC | 5,952 | Claims | Covid-19 | Brazil, Indonesia, Nigeria | Eng., pt., id. | 2020 | 2022 |
| ESOC Covid-19 | 5,613 | Claims | Covid-19 | International | 35 languages | 01/01/2020 | 1/12/2020 |
| FakeCovid | 7,623 | Claims | Covid-19 | International | 40 languages | 04/01/2020 | 01/07/2020 |
| FibVID | 1,353 | Claims | Covid-19 | International | English | 02/2020 | 01/2021 |
| WICO | 364,325 | Claims | Covid-19 | – | English | 17/01/2020 | 30/06/2021 |
| Twitter16 | 818 | Claims | Various | – | English | 03/2015 | 12/2016 |
| Rumors | 1,022 | Claims | Various | USA, UK, China, India | English | 01/05/2017 | 01/11/2017 |
| ClaimsKG | 74,066 | Claims & Knowledge Graph | Various | International | English | 1996 | 2023 |
| IFND | 56,868 | Claims & Images | Various | India | English | 2013 | 2021 |
| Verite | 1,001 | Claims & Images | Various | International | English | 01/01/2001 | 01/01/2023 |
| LIAR | 12,836 | Claims | Politics | USA | English | 2007 | 2016 |
| LIAR-New | 1,957 | Claims | Politics | USA | English & French | 10/2021 | 11/2022 |
| TruthSeeker2023 | 180,000 | Claims | Politics | USA | English | 2009 | 2022 |
| Check-COVID | 1,504 | Claims | Covid-19 | – | English | – | – |
| Climate-Fever | 7,675 | Claims | Climate | – | English | – | – |
| CMU-MisCOV19 | 4,573 | Claims | Covid-19 | – | English | – | – |
| COVID-Fact | 4,086 | Claims | Covid-19 | – | English | – | – |
| FaVIQ | 188,376 | Claims | Various | – | English | – | – |
| FEVER | 185,445 | Claims | Various | USA | English | – | – |
| FEVEROUS | 87,026 | Claims | Various | – | English | – | – |
| HoVer | 26,171 | Claims | Various | – | English | – | – |
| MediaEval | 15,629 | Claims & Images | Various | – | English, Spanish | – | – |
| MM-COVID | 11,173 | Claims | Covid-19 | International | 6 languages | – | – |
| PHEME | 62,445 | Claims | Newsworthy events | International | English | – | – |
| PubHealthTab | 1,942 | Claims | Health | North America | English | – | – |
| Snopes Fact-news | 4,550 | Claims | Various | USA | English | – | – |
| Twitter15 | 1,490 | Claims | Various | – | English | – | – |
| X-Fact | 31,189 | Claims | Various | – | 25 languages | – | – |
| DeFaktS | 105,855 | Claims | Various | Germany | German | – | – |
| MultiClaim | 31,305 | Claims | Various | International | 39 languages | – | – |
| NLP4IF-2021 | 3,172 | Claims | Covid-19 | International | Ar., bul., eng. | – | – |
| Benjamin Political News | 296 | News articles | Election | USA | English | 02/2016 | 11/2016 |
| BuzzFeedNews | 2,282 | News articles | Election | USA | English | 19/09/2016 | 27/09/2016 |
| CT-FAN | 2462 | News articles | Various | Germany, USA, Canada | English & German | 2010 | 2022 |
| Fake News Elections | 38,333 | News articles | Politics | USA | English | 04/2023 | 10/2023 |
| FakeNews | 486 | News articles | Politics | USA | English | 01/2016 | 10/2017 |
| FA-KES | 804 | News articles | Syrian war | Syria | English | 2011 | 2018 |
| FANG-COVID | 41,242 | News articles | Covid-19 | Germany | German | 02/2020 | 03/2021 |
| ISOT Fake News | 44,898 | News articles | Politics | International | English | 2016 | 2017 |
| Italian disinformation | 16,867 | News articles & Tweets | Election | Italy | English, Italian | 01/01/2019 | 27/05/2019 |
| Med-MMHL | 40,601 | News, tweets, images & LLM-generated | Health | USA | English | 01/01/2022 | 01/05/2023 |
| NELA-GT-2020 | 1,779,127 | News articles & Tweets | Various | USA | English | 01/01/2020 | 31/12/2020 |
| ReCOVery | 142,849 | News articles & Tweets | Covid-19 | – | English | 01/2020 | 05/2020 |
| Spanish Fake News Corpus | 572 | News articles & Social media posts | Various | International | Spanish | 01/11/2020 | 31/03/2021 |
| Weibo21 | 9,128 | News articles | Various | China | Chinese | 12/2014 | 03/2021 |
| BanFakeNews | 50,000 | News articles | Various | Bangladesh | English | – | – |
| Celebrity | 500 | News articles | Celebrities | – | English | – | – |
| COVID-19 Fake News | 10,700 | News articles & Social media posts | Covid-19 | – | English | – | – |
| Fact-check-tweet | 13,070 | News articles & Tweets | Various | International | 4 languages | – | – |
| FakeHealth | 440,870 | News articles & Social media posts | Health | USA | English | – | – |
| FakeNewsAMT | 480 | News articles | Various | – | English | – | – |
| FakeNewsCorpus | 9,408,908 | News articles | Various | – | English | – | – |
| FakeNewsNet | 23,196 | News articles | Politics & Celebrities | – | English | – | – |
| FNC-1 | 49,972 | News articles | Various | – | English | – | – |
| Misinfo Reaction Frames | 25,100 | News articles | Global crises | International | English | – | – |
| MuMin | 21,565,018 | News articles, Tweets & Images | Various | International | 41 languages | – | – |
| TI-CNN | 20,015 | News articles | Politics | USA | English | – | – |
| BuzzFace | 1,176,713 | Social Media posts | Election | USA | English | 01/09/2016 | 30/09/2016 |
| FacebookHoax | 15,500 | Social Media posts | Hoaxes | – | English | 01/07/2016 | 31/12/2016 |
| FACTOID | 3,354,450 | Social Media posts | Politics | USA | English | 01/2020 | 04/2021 |
| Fakeddit | 1,063,106 | Social Media posts & Images | Various | – | English | 19/03/2008 | 24/10/2019 |
| VoterFraud2020 | 7,600,000 | Social Media posts, Images & Videos | Election | USA | English | 23/10/2020 | 16/12/2020 |
| MR2 | 14,700 | Social Media posts & Images | Rumor | USA & China | English & Chinese | – | – |
| Reddit | 12,597 | Social Media posts | Various | USA | English | – | – |
| DBpedia | 1,950,000 | Wikipedia text | Various | International | 14 languages | – | – |
| ICWSM | 2,500 | Images | Election | Brazil & India | 10 languages | 10/2018 | 06/2019 |
| FaceForensics++ | 1,800,000 | Images | Deepfakes | – | – | – | – |
| FCV-2018 | 380 | Videos | Various | International | 5 languages | 2016 | 2017 |
| Celeb-DF | 5,639 | Videos | Celebrities | – | – | – | – |
| DEEPFAKETIMIT | 640 | Videos | Various | – | – | – | – |

Twitter16, and PHEME datasets. However, in both of these studies, the limited number of datasets analyzed fails to provide a comprehensive view of the diverse landscape of available datasets in this field. Similarly, Pelrine et al. (2023) highlights issues of ambiguous claims in the LIAR dataset, but does not expand their analysis beyond that one and their own LIAR-New dataset.

In short, existing works often only briefly discuss the structure and the content of datasets when addressing data issues, frequently lacking detailed analysis or focusing on a limited number of cases. To overcome this problem, we present one of the most comprehensive surveys of misinformation datasets to date by analyzing their overall content and potential effectiveness in detecting false information.

## 3 SURVEY

In this section, we introduce a collection of combined datasets on true and false information, including a total of 75 datasets, with 35 specifically focused on claims and statements. The full dataset contains a total of 120,901,495 observations, while the subset that we further analyze includes

1,612,933 observations. These data encompass a wide range of topics, including political issues, health concerns, and environmental questions, often related to the United States but also covering international news, headlines and online posts. The original labels within the datasets were assigned through a combination of expert evaluations and algorithmic methods. The following section provides a detailed summary of the data collection process and the characteristics of these datasets.

## 3.1 COLLECTION PROCESS

Our data collection process involved an exhaustive search of journal and conference articles to identify relevant datasets. To achieve this, we used the Google Scholar search engine with keywords such as "fake news", "disinformation", "misinformation", "dataset", "detection", "survey", and others highlighted in Appendix A.1. We focused on articles published between 2016 and 2024. This initial phase allowed us to collect 28 datasets.

We then expanded our selection by rigorously examining the citations in scientific articles related to these initial datasets. Some articles listed available datasets in their literature reviews or surveys, enabling us to incorporate additional data. Through these combined approaches—which are further detailed in the Appendix—we identified 75 publicly accessible datasets, presented in Table 1.

For our analyses, we refined our selection to focus exclusively on datasets containing textual claims, defined here as short statements ranging from one to two sentences. Tweets are included in this definition, while lengthier online and social media posts are excluded. We chose to focus on this type of data because statements and claims are more concise than other forms of information, such as OP-ED and news articles, which often include opinions, commentary, and contextual details. This extraneous information can obscure the core claim or statement and introduce noise in the labeling process, as information can be partly true or false. Therefore, to test label quality, we initially focus on claims and statements to increase the reliability of our findings. This approach paves the way for future research to investigate the labeling of other types of content. Thus, with this filtering criteria, we reduced our selection to 35 datasets.

## 3.2 CLAIMS DATASETS

A summary description of each of our datasets can be found in Appendix A.2. Of these datasets, 12 consist of claims scraped from fact-checking or reliable websites, another 12 consist of tweets, and the remaining 11 comprise claims drawn from Twitter, the internet, social media, or news websites. There is variation in the topics of these datasets, but most focus on areas with significant societal impact where misinformation is prevalent and potentially harmful. For example, 16 of the datasets focus on health, vaccination, and COVID-19; 3 focus on political issues; 1 on environmental issues, and the rest covers various subjects, from culture, sport, the economy and so on. Unfortunately, a significant limitation of much of this data is the absence of information regarding the date the claim was made or fact-checked. This can potentially impact the accuracy of labeling, given that certain claims may have been true or false at the time they were made. In addition, this limitation affects the scope of our temporal leakage analysis. Consequently, scholars, and practitioners alike should be cautious when using these data.

## 4 DATA QUALITY

Various factors affect the validity of the labels, which in turn impacts the overall quality of the data. In this section, we first discuss the strengths and weaknesses of each labeling technique used and discuss how we standardized the labels across studies. Next, we evaluate whether certain keywords are spuriously predictive of the veracity of the claims, and, finally, we examine if the datasets suffer from spurious temporal correlations.

### 4.1 LABELING APPROACH

The task of annotating statements is both crucial and challenging for anyone attempting to train a robust classifier for disinformation detection. Precise labeling is essential to ensure the classifier's effectiveness, as it directly impacts its performance and reliability. Numerous approaches have been proposed in the literature to label true and false information. These approaches include expert and

Table 2: Labelling approach and distribution for 35 claim datasets (subset of Table 1).

| Dataset | Labeling approach | True claims (%) | False claims (%) | Other (%) |
|---|---|---|---|---|
| AntiVax | Human expert | 38.15 | 61.85 | 0 |
| Check-COVID | Human expert | 37.92 | 16.48 | 24.92 |
| Climate-Fever | Human expert | 52.64 | 16.48 | 30.88 |
| CMU-MisCOV19 | Human (N.S.) | 17.75 | 74.74 | 7.51 |
| CoAID | Source-based categorization (T) & Human expert (F) | 93.48 | 6.52 | 0 |
| Counter-covid-19-misinformation | Human (N.S.) | 0 | 100 | 0 |
| COVID-19-Rumor | Human expert | 26.16 | 73.84 | 0 |
| Covid-19-disinformation | Crowd-sourced | 86.32 | 13.68 | 0 |
| COVID-Fact | Human-expert (T) & Algorithm-generated creation (F) | 31.72 | 68.28 | 0 |
| Covid-vaccine-misinfo-MIC | Crowd-sourced | 56.71 | 33.61 | 9.68 |
| DeFaktS | Human expert | 58.86 | 41.14 | 0 |
| ESOC Covid-19 | Human expert | 0 | 100 | 0 |
| FakeCovid | Human expert | 0.86 | 94.99 | 4.15 |
| FaVIQ | Algorithm & Validation by human | 49.77 | 50.23 | 0 |
| FEVER | Crowd-sourced | 54.92 | 20.46 | 24.62 |
| FEVEROUS | Algorithm | 57.77 | 38.77 | 3.46 |
| FibVID | Human expert | 23.76 | 76.24 | 0 |
| HoVer | Crowd-sourced | 58.74 | 41.26 | 0 |
| IFND | Human expert | 66.65 | 33.35 | 0 |
| LIAR | Human expert | 55.67 | 44.33 | 0 |
| LIAR-New | Human expert | 15.02 | 84.98 | 0 |
| MediaEval | Source-based categorization | 46.05 | 53.95 | 0 |
| MM-COVID | Human expert | 71.74 | 28.26 | 0 |
| MultiClaim | Human expert | 72.14 | 27.86 | 0 |
| NLP4IF-2021 | Crowd-sourced | 95.15 | 4.85 | 0 |
| PHEME | Human expert and non-expert | 22.14 | 77.86 | 0 |
| PubHealthTab | Human expert | 52.47 | 23.79 | 23.74 |
| Rumors | Algorithm | 13.78 | 70.47 | 15.76 |
| Snopes Fact-news | Human expert | 16.93 | 63.57 | 19.50 |
| TruthSeeker2023 | Crowd-sourced | 51.36 | 48.64 | 0 |
| Twitter15 | Human expert | 50.07 | 24.83 | 25.10 |
| Twitter16 | Human expert | 50.37 | 25.06 | 24.57 |
| Verite | Human expert | 33.77 | 66.23 | 0 |
| WICO | Human expert | 30.83 | 60.76 | 8.41 |
| X-Fact | Human expert | 63.63 | 33.40 | 2.97 |

(T) indicates the method used to establish true claims
(F) indicates the method used to determine false claims
(N.S.) indicates that expertise is not specified

crowd-sourced annotation, source-based techniques, algorithmic methods, and a hybrid of these different approaches, all of which have been used in at least one of our 35 datasets (see Table 2). We describe these different approaches in turn to highlight their potential advantages and limitations.

**Expert-based approach**   Experts and fact-checkers are a small group of non-partisan professionals from various disciplines who manually verify the veracity of information. The result of these verifications are often published in fact-checking websites such as *Politifact* or *Snopes*. The strength of this approach lies in its rigorous review process, ensuring each piece of information is thoroughly evaluated, which leads to consistent reviews across fact-checkers. However, this method is not scalable and is costly (Zhou & Zafarani, 2020). As a result, experts must selectively choose the information they evaluate, which leads to many pieces of information going unchecked and potential biases in the selection of news and information that is evaluated (Lee et al., 2023; Markowitz et al., 2023; Walker & Gottfried, 2019).

**Crowd-sourced approach**   Crowdsourced fact-checking involves enlisting non-expert laypeople to assess the accuracy of online information. These evaluations are then aggregated to determine the veracity of the content. This approach is advantageous because it is more scalable, and laypeople can respond to misinformation much more quickly than professional fact-checkers (Zhao & Naaman, 2023). Additionally, this method has been shown to be effective in reducing the spread of misinformation and to produce veracity ratings similar to those of professional fact-checkers (Allen et al., 2021; Martel et al., 2024). However, crowdsourcing also has its limitations. It can be challenging to filter out evaluations from non-credible users and to ensure a balanced representation of users from different partisan backgrounds (Zhou & Zafarani, 2020; Martel et al., 2024).

**Source-based approach**    Source-based approaches to verifying information involve evaluating the domain or author of the content. Information is then rated as accurate if it comes from reliable sources and inaccurate otherwise. This method is more scalable than manual fact-checking, as it consists of evaluating the credibility of the source rather than each individual story. Additionally, this method is proven to be reliable, as experts generally rate news domains similarly (Lin et al., 2023). However, there are notable drawbacks. For instance, individual stories can vary in accuracy even within the same source, and not all content from low-quality outlets is necessarily false or misleading. Additionally, source familiarity significantly influences the perceived trustworthiness of content. Sources that are unfamiliar are often less trusted, which can lead to unfair negative evaluations of high-quality but lesser-known sources (Pennycook & Rand, 2019; Williams-Ceci et al., 2023)

**Algorithmic methods**    Finally, algorithmic methods can also be used to evaluate the veracity of content using NLP or other ML techniques (Zhou & Zafarani, 2020). For example, Covid-fact uses a BERT-based classifier, FaVIQ uses T5-3B, and Rumors uses an approach based on a social graph. These methods offer significant advantages in scalability, as they can process vast amounts of data quickly and efficiently, making them suitable for large-scale verification tasks. However, their accuracy can be questionable in many cases, ranging from struggles with nuanced or context-specific content (Boukouvalas & Shafer, 2024), issues with transfer and generalization (Huang et al., 2020; Pelrine et al., 2021; 2023), or just generically poor performance (e.g., even state-of-the-art methods often have below 70% accuracy compared to human labels (Zhang & Gao, 2023; Pelrine et al., 2023)). Thus, the quality of algorithmic labels is often dubious.

**Our label unification approach**    In this survey, we use the original labels from the studies to create a more consistent categorical variable across datasets. This allows us to test the accuracy of the veracity of the labels in each dataset. Specifically, we classify content as true, false, or unknown. Although this categorical variable is less precise than the detailed scales used to classify some of the claims in the data, we adopted this approach to ensure consistency across all datasets. Information that is mostly true is classified as "true", while content that is mostly false or hyperpartisan is classified as "false". This method aligns with existing research indicating that people respond similarly to content that is false or hyperpartisan (Ross et al., 2021; Pennycook et al., 2020). Finally, claims that could not be verified or were ambiguous, such as those partially true or false, were classified as "unknown". The percent of true and false claims in each dataset using this coding scheme is shown in Table 2.

## 4.2 SPURIOUS KEYWORD CORRELATIONS

We next evaluate whether there are certain keywords that overpredict misinformation in the claim datasets. We adapt the approach that Pelrine et al. (2021) used to check for spurious temporal correlations. Specifically, we trained a random classifier with the 40 most frequent words in each dataset, after removing stop-words. Utilizing scikit-learn, we set a maximum tree depth of 20 and retained the other default settings. The macro F1 results are shown in Table 3.

We particularly flag five datasets for spurious correlations between certain words and labels: IFND, MM-COVID, TruthSeeker2023, CoAID, and Twitter16. For example, consider Truthseeker2023. Nearly all tweets here mentioning politicians are labeled as "false", with only those containing "Trump" showing more variation (see also Appendix A.3). Obviously, in the real world, tweets mentioning politicians are obviously not exclusively false. Thus, models trained on data like Truthseeker2023 risk generalizing inaccurate results, and doing so on topics extremely sensitive to bias like discussion of politicians. Moreover, these findings extend beyond political names. Terms like "michigan", "vaccines", "immunity", "pfizer", and so on are consistently labeled as false, while words like "marijuana", "wealth", "terrorism", "radical" and others are always associated with the true label (see also Figure 2 in the Appendix). Similar patterns are also observed in the other four datasets with highest keyword predictivity. Therefore, we urge caution about training and testing models on these datasets, especially text-focused ones.

## 4.3 SPURIOUS TEMPORAL CORRELATIONS

Pelrine et al. (2021) highlighted how collecting data of different classes at different times can make temporal information unrealistically predictive. For example, discussion of particular news events can become excessively correlated with veracity labels, leading to classifiers that rely on these events

Table 3: Keywords correlations evaluation. A high score means that the keywords provide an unrealistically strong prediction.

| Dataset | Keywords Predictivity (% F1) |
|---|---|
| Check-COVID | 22.9 |
| Climate-Fever | 23.0 |
| CoAID | 60.9 |
| COVID-19-Rumor | 37.5 |
| COVID-FACT | 40.6 |
| DeFaktS | 37.1 |
| FakeCovid | 33.3 |
| FaVIQ | 34.8 |
| FEVER | 22.9 |
| FEVEROUS | 24.4 |
| FibVID | 53.3 |
| HoVer | 38.1 |
| IFND | 82.2 |
| LIAR | 44.7 |
| LIAR-New | 46.0 |
| MM-COVID | 77.1 |
| MultiClaim | 41.9 |
| NLP4IF-2021 | 48.8 |
| PubHealthTab | 31.0 |
| Rumors | 27.6 |
| Snopes Fact-news | 25.9 |
| TruthSeeker2023 | 66.8 |
| Twitter15 | 35.7 |
| Twitter16 | 55.2 |
| Verite | 34.0 |
| X-Fact | 28.8 |

The table excludes datasets with tweet IDs and single veracity labels.

having artificially inflated performance, that will not generalize to the real world where veracity cannot be determined by past events alone.

Like in the preceding section, we follow their proposal for evaluating datasets for this limitation, training a random forest classifier. As feature, we use either the first three digits of the tweet ID (which contain time information) as in Pelrine et al. (2021) for Tweet datasets, or the date itself for datasets which include it. For the latter, we encode it as the integer number of days since the first date in the dataset. We exclude from this analysis datasets without either form of temporal information.

Table 4: Temporal correlations evaluation. A high score here means time—and information correlated with it—is unrealistically predictive.

| Dataset | Evaluation Type | Temporal Predictivity (% F1) |
|---|---|---|
| CoAID | Date | 48.3 |
| FakeCovid | Date | 49.7 |
| FibVID | Date | 62.2 |
| LIAR-New | Date | 53.7 |
| Rumors | Date | 69.1 |
| X-Fact | Date | 63.2 |
| AntiVax | TweetID | 46.8 |
| CMU-MisCOV19 | TweetID | 45.6 |
| Covid-19-disinformation | TweetID | 46.5 |
| MediaEval | TweetID | 72.2 |
| Twitter15 | TweetID | 83.2 |
| Twitter16 | TweetID | 96.5 |
| WICO | TweetID | 40.7 |

Results are shown in Table 4. We first note that our findings on Twitter15 and Twitter16 are similar to Pelrine et al. (2021), confirming these datasets have extreme issues with spurious correlations in temporal information. They should not be used without carefully and explicitly addressing this limitation. While not as severe, we also see that MediaEval and Rumors also suffer from some significant spurious temporal correlations, and caution is advised. The rest of the datasets have a substantially better temporal balance, with the temporal feature offering little better than random

Table 5: State-of-the-art GPT-4 baselines, with and without web search.

| Dataset | F1 (Search) | F1 (Offline) |
|---|---|---|
| IFND | 56.5% | 42.0% |
| checkcovid | 78.8% | 85.4% |
| climate_fever | 66.9% | 65.3% |
| coaid | 62.0% | 60.3% |
| covid_19_rumor | 62.8% | 65.8% |
| covidfact | 67.5% | 67.4% |
| fakecovid | 50.4% | 51.0% |
| faviq/test | 81.5% | 80.7% |
| fever/paper_test | 88.6% | 89.2% |
| feverous/validation | 65.6% | 62.2% |
| fibvid | 67.6% | 67.3% |
| hover/validation | 68.8% | 61.7% |
| liar/test | 44.8% | 50.7% |
| liar_new | 69.7% | 63.6% |
| mm-covid | 85.6% | 86.5% |
| pheme | 34.3% | 33.4% |
| pubhealthtab/test | 30.8% | 49.3% |
| rumors | 69.5% | 80.7% |
| snopes | 90.6% | 81.4% |
| truthseeker2023 | 81.9% | 81.0% |
| twitter15 | 57.7% | 66.5% |
| twitter16 | 49.2% | 55.8% |
| verite | 63.3% | 59.8% |
| x_fact/test | 55.0% | 53.0% |

performance. However, we note that only a small fraction of the total datasets include dates, and recommend that future datasets add this important information.

### 4.4 PARTISANSHIP AND IDEOLOGICAL LEAN

We conducted analyses using large language models (LLMs) to assess the partisan and ideological lean of claims within various datasets. By providing distributions of these leans, we offer an approach for selecting datasets that better align with specific research goals. We also aim to determine if the veracity of a statement is spuriously correlated with its partisan lean, because such an association could in turn lead to models inaccurately assessing truthfulness based on that lean alone. Our findings indicate that a slightly higher proportion of true statements across the datasets are predicted to lean Democratic (12.12%) compared to those leaning Republican (9.5%). In contrast, a larger proportion of false statements are predicted to lean Republican (19.44%) than Democratic (9.36%). However, these proportions vary significantly across datasets, highlighting the importance of careful dataset selection. For a full table of results and further analysis, please see Appendix A.4.

### 5 EVALUATION

#### 5.1 BASELINE PERFORMANCE

We next discuss baselines when using these datasets. Recent works have shown LLMs represent the state-of-the-art for misinformation detection on generic claims Chen & Shu (2023); Pelrine et al. (2023). But despite its importance Pelrine et al. (2021), both human and compute time constraints can be a barrier to comparing with such strong baselines.

Thus, we provide two baselines for future use. We follow the recent method of Tian et al. (2024) and use GPT-4-0125 in two ways: directly prompting the LLM for a veracity evaluation, and providing the LLM a web search tool to first collect evidence before forming a final verdict. We note that although these are state-of-the-art systems for zero-shot misinformation detection, they should not be

regarded as sole or permanent points of comparison. Stronger LLMs and methods could replace them eventually. Nonetheless, they can provide a useful point of comparison for the near future.

We note that 8 datasets are excluded from this baseline: 7 tweet datasets that we were unable to retrieve due to X API limits, and the ESOC Covid-19 dataset because it only has a "refutes" label. Results on all others are provided in Table 5. Notably, because these are zero-shot approaches, they are much less vulnerable to spurious correlations than models trained on each of these datasets, sometimes leading to dramatically lower but more realistic performance compared to alternatives in the literature (e.g., Twitter15 and Twitter16, where temporal classification achieves over 80% F1).

## 5.2 THE FLAW IN CURRENT METRICS

When looking at the outputs of the prediction system, we observe cases where the predicted label did not match the ground truth, yet the evidence and reasoning of the system was valid. For instance, in one example on the FEVER dataset, the input claim is "Vietnam is a place" and the prediction said roughly "Vietnam is not just a place, it's a country!" In another example from LIAR-New, a statement was marked false by PolitiFact because it was in the context of a fake video, but the statement itself did not mention the video and in isolation would be true. In cases like these (and further examples in Appendix A.6), a simple binary or categorical label cannot provide an informative evaluation.

To determine the prevalence of this phenomenon, two annotators manually evaluated (Appendix A.7) chain-of-thought rationales from the web-search enabled baseline prediction system Tian et al. (2024).

Table 6: Agreed-upon manual annotations and inter-annotator agreement. Many examples marked invalid by categorical labels are actually valid.

| Dataset | Label ≠ Prediction | | Label = Prediction | |
|---|---|---|---|---|
| | Rationale is Valid | Rationale is Invalid | Rationale is Valid | Rationale is Invalid |
| LIAR-New | 55/100 | 30/100 | 76/100 | 1/100 |
| FEVER | 38/100 | 34/100 | – | – |
| MM-COVID | 39/70 | 3/70 | 89/100 | 0/100 |

We observed a consistently high false-incorrect rate (first column of Table 6) and a generally low false-correct rate (fourth column of Table 6). Therefore, when benchmarking generative AI misinformation detection systems using categorical labels, the predictive accuracy and similar metrics reflect a reasonable lower bound on the performance—but a terrible upper one. We also observe that there is a large amount of ambiguity and room for interpretation in the examples that are being marked wrong by categorical label in these three datasets. Hence:

1. Categorical metrics cannot be used alone to compare generative and non-generative systems. Although multiple recent works (e.g., Pelrine et al. (2023); Chen & Shu (2023); Wei et al. (2024); Yu et al. (2023)) have highlighted the effectiveness of recent LLMs for misinformation detection, their comparisons with prior approaches may even still be underestimating the dominance of LLMs in this domain.
2. Generative systems need many, repeated, and large-margin measurements if the categorical lower bound alone is to form meaningful comparisons between them.
3. There is an urgent need for better datasets and better evaluation procedures in this domain that are suitable for the generative AI era.

Although to our knowledge not addressed in the context of misinformation detection on claims, challenges in evaluation of generative AI have been broadly documented in other fields (McIntosh et al., 2024; Ahuja et al., 2023; Michel-Villarreal et al., 2023; Basole & Major, 2024). A common approach aimed at solving this is using an LLM for evaluation (Bai et al., 2022; Sun et al., 2023b), with Sun et al. (2023a) noting that LLM-powered evaluation can produce more consistent preference signals than human annotators. In general, using LLMs for evaluation enables one to leverage much richer signals than simple categorical predictions and labels, while avoiding reliance on often inaccessible human evaluators.

As an initial step towards higher-fidelity evaluation, we constructed an evaluator based on contradictions between the explanation generated by a predictive system, and a fact-checking article. In

particular, we provide GPT-4-0409 both the prediction and the article, and ask it to score contradictions from 0 (none) to 10. The exact prompt and other implementation details are provided in Appendix A.8. We chose a score-based approach to avoid forcing a potentially misleading binary in cases where there is a partial contradiction. With this approach, good predictions should have low contradiction against a high quality, professional fact-checking article. We also tested binary and trinary versions of this prompt, described in the Appendix, which yielded nearly identical results.

Specifically, with the oracle-optimal threshold of 3 or less indicating a prediction that is not wrong, and 4 or more indicating one that is wrong, this evaluation agrees 68% of the time with the human labels of wrong and not wrong predictions on the LIAR-New dataset. This is higher than the 60% original human agreement on this dataset (before disagreement resolution described in Appendix A.7), and suggests the method extracts a meaningful but not definitive evaluation signal. We also note, though, that there is more to high quality misinformation detection than just a lack of contradiction. Therefore, we do not suggest using this tool as a primary evaluator. But we provide these results, along with all the data—inputs, predictions, manual labels, fact-checking articles, and evaluator outputs—on our GitHub, as a potential springboard for stronger evaluation methods in future research.

## 6 LIMITATIONS

First, we note that while to our knowledge this represents the largest and most comprehensive survey of datasets in this domain, there are certainly many other datasets in existence and it is probable that some were not included. There is also a steady stream of new datasets being created. In the near future, we plan to collect external feedback and update our survey to maintain and expand the comprehensiveness of our study.

We also note that our unified label schema simplifies some labels that might have meaningful information, for example, gradations of veracity instead of binary true/false. Some granularity has been traded for the ability to establish a unified schema across all the claims datasets. When using these datasets, we advise careful consideration of the optimal labels to apply.

As discussed previously, additional work is needed in evaluation, both to confirm that the observed validity issues with metrics like accuracy are widespread (as we hypothesize) and to create strong, thoroughly tested alternatives. We also note that the baselines we have provided use old evaluation procedures on LLM-based predictors. This can be flawed both for the reasons discussed in Section 5.2, and potentially also because a substantial proportion of the data could be within the LLM training data. Pelrine et al. (2023) indicates LLM-based methods offer the strongest performance even beyond their knowledge cutoffs, and using web search to actually provide evidence can mitigate this to some degree. But nonetheless, these baselines should be viewed carefully and with due attention to both their strengths and limitations, and future work to establish more universal baselines—as well as datasets and evaluation methods that enable them—would be very valuable.

Lastly, although we discussed multiple key aspects of misinformation detection datasets, there are still more that are worth considering. For example, much of our analysis focuses on claims datasets, which are unquestionably important but by no means the entirety of valuable data in the field. Similarly, although dataset quality in terms of labels and spurious correlations is critical, there can be other important considerations like ambiguous statements (Pelrine et al., 2023). We aim to address some of these in future work.

## 7 CONCLUSION

High quality data is essential for realistic results and rapid progress in the field. In this work, we have provided a guide to misinformation detection datasets aiming at both quantity and quality. It also highlights limitations of existing datasets and evaluation approaches, which may have uncertain labels, spurious correlations, and misleading results. We hope that on the one hand this work can provide a roadmap for future methods research that needs to select datasets and evaluation approaches, and on the other, provide the foundational understanding and call to action to improve the misinformation detection dataset landscape.

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

# A APPENDIX

## A.1 SUPPLEMENT ON DATA COLLECTION

For the dataset search, all of the following set of words, as well as the ones presented in Section 3.1, were used in Google Scholar: "fake news", "false news", "fake news dataset", "false news dataset", "fake news database", "false news database", "misinformation dataset", "misinformation database", "misinformation detection", "misinformation survey", "disinformation dataset", "disinformation database", "disinformation detection", "disinformation survey", "fact check dataset", "fact check database", "benchmark for fake news detection", "benchmark dataset for fake news", "misinformation data", "dataset for evidence-based fact-checking", "fact-checking corpus", "fact verification corpus", and "misinformation detection review". In addition to these terms, and as mentioned previously, a year filter was used using the advanced search. Only articles dated between 2016 and 2024 were included.

Once the initial datasets were identified, we then expanded our selection by (1) identifying the most frequently cited papers related to these datasets (based on the number of citations in Google Scholar) and (2) carefully reviewing these papers to uncover additional dataset. This review process primarily focused on analyzing the articles' literature reviews and reference lists to identify datasets that were mentioned and could be pertinent to our research. For instance, according to Google Scholar, the article by Shu et al. (2020), which introduces the FakeNewsNet dataset, has been cited 1,190 times, ranking it among the top four most frequently cited articles that we collected. Based on this, we proceeded to review *Section 2* of the article, titled *Background and Related Work*. In this section, the authors mention six existing datasets for misinformation detection: BuzzFeedNews, LIAR, BS Detector, Credbank, BuzzFace, and FacebookHoax. If we had not already gathered these datasets during our initial keyword search on Google Scholar, we collected them at this stage. We also maintained the same publication year criterion, considering only datasets published between 2016 and 2024. Consequently, Credbank, which was published in 2015 in ICWSM'15, was excluded. BS Detector was no longer publicly available.

## A.2 DATASET DETAILS

This section provides an overview of datasets containing claims. The number of claims, the collection method, and the original labels are discussed.

**AntiVax** (Hayawi et al., 2022): AntiVax is a dataset containing 15,465,687 tweets about the COVID-19 vaccine, of which only 15,073 are annotated for the model training. These were collected via the Twitter API form December 1, 2020 to July 31, 2021. The annotations are binary (misinformation or not misinformation). Tweets labeled as misinformation include opinions or general news about the vaccine. Tweets containing sarcasm or humor are not classified as misinformation.

**Check-COVID** (Wang et al., 2023): This dataset contains 1,504 expert-annotated claims about the COVID-19 pandemic. These claims are either composed by annotators or extracted from news articles. Each claim is also paired with a sentence evidence from scientific journals. Labels are divided into three categories: support, refute or not enough info.

**Climate-Fever** (Diggelmann et al., 2020): Climate-Fever is a dataset about climate change. It includes 7,675 annotated claim-evidence pairs. Claims are collected on the Internet while evidences are retrieved from Wikipedia. Each claim is assigned one of the following labels: supports, refutes or disputed.

**CMU-MisCOV19** (Memon & Carley, 2020): CMU-MisCov19 is a dataset about COVID-19. It contains tweets that were collected over three days: March 28, 2020, June 15, 2020, and June 24, 2020. 4,573 tweets are annotated based on various types of information and misinformation. In total, there are 17 categories, such as irrelevant, conspiracy, true treatment, fake cure, false fact, ambiguous, etc.

**CoAID** (Cui & Lee, 2020): This dataset covers various COVID-19 healthcare misinformation. It contains 4,251 news, 926 social platforms posts, and 296,000 related user engagements. All facts are collected between December 1, 2019 and September 1, 2020. All the data is annotated in a binary form: true or fake.

**Counter-covid-19-misinformation** (Micallef et al., 2020): Covering four-month period, this dataset contains 155,468 tweets relating to COVID-19 and, more specifically, fake cures and 5G conspiracy. The tweets were harvested from an existing dataset [3] and Twitter. 4,800 claims are annotated, and the labels are divided into three categories: misinformation, counter-misinformation, or irrelevant.

**COVID-19-Rumor** (Cheng et al., 2021): This dataset includes 7,179 annotated claims crawled from Google and Twitter from January 2020 to March 2020. The topics of these claims, all related to COVID-19, include emergency events, comments from public figures, updates on the coronavirus outbreak, etc. The labels were manually assigned and cross-validated. The labels are also divided into three categories, consisting of true, false, or unverified.

**Covid-19-disinformation** (Alam et al., 2020): This is another dataset about COVID-19 disinformation. It contains 16K coded claims in Arabic, Bulgarian, Dutch, and English. These were collected via the Twitter API between January 2020 and March 2021. Their labels are fined-grained. The annotation task involved determining the truthfulness of the tweet, its potential to cause harm, whether it is relevant for policymakers, etc.

**COVID-Fact** (Saakyan et al., 2021): Also on the subject of COVID-19, Covid-Fact contains 4,086 claims. Among these, 1,296 are factual claim from the $r/COVID19$ subreddit, while 2,790 are false claims automatically generated. All claim contain evidence, and the labels are binary: supported or refuted.

**Covid-vaccine-misinfo-MIC** (Kim et al., 2023): Covid-vaccine-misinfo-MIC is a geolocated and multilingual dataset about COVID-19. It spans from 2020 to 2022, and includes 5,952 tweets from Brazil, Indonesia, and Nigeria. The claims are all labeled in a granular form, indicating whether they are vaccine-related, contain misinformation, are political, etc.

**DeFaktS** (Ashraf et al., 2024) : DeFaktS is a database of 105,855 claims from X (formerly Twitter), of which 20,008 are annotated. Claim topics are varied. They include, for example, war in Ukraine, elections, covid-19 pandemic, energy crisis, climate, inflation, etc. All the claims are written in German and the veracity labels are fine-grained, as they include binary labels (real, fake) and labels stating content, authenticity, psychology and semantic features.

**ESOC Covid-19** (Siwakoti et al., 2021): ESOC contains 5,613 claim-stories about misinformation gathered from the early days of the COVID-19 pandemic up to the end of December 2020. These claims come from all five continents and all contain misinformation.

**FakeCovid** (Shahi & Nandini, 2020): FakeCovid is a dataset containing news claims about COVID-19. These data were collected from 92 different fact-cheking websites between January 4, 2020, and May 15, 2020, covering 40 languages and originating from 105 countries. The truthfulness labels (false, mostly false, misleading, half true, mostly true, no evidence) are derived from experts at fact-checking agencies. The dataset also includes other labels defining the type of false news (prevention & treatments, international response, conspiracy theories, etc), all annotated by members of their team.

**FaVIQ** (Park et al., 2021): This dataset contains 188K annotated claims and evidences. Each claim has been converted based on questions from the Google Search queries. The claims cover various subjects including culture, sports, and history. The labels are binary: support or refute.

**FEVER** (Thorne et al., 2018): This dataset includes 185,445 coded claims generated by altering sentences extracted from the 50,000 most popular Wikipedia pages. Annotators were tasked with crafting claims covering a wide array of topics, ranging from historical facts to entertainment trivia, each containing a single fact. The labels assigned to these claims were determined based on evidence sourced from Wikipedia as well, and they were categorized in a binary manner as either supported or refuted.

**FEVEROUS** (Rami et al., 2021): Continuing in the same vein as FEVER, FEVEROUS is a dataset containing 87,026 claims extracted from Wikipedia. Each claim is annotated based on associated evidence. One distinctive feature with FEVER is that the labels are divided into three categories: supported, refuted, or not enough information.

---

[3]https://doi.org/10.2196/19273

**FibVID** (Kim et al., 2021): This COVID-19 related dataset was collected by crawling 1,353 news claims and the labels of two fact-checking websites, Politifact and Snopes. These news claims were subsequently matched with 221,253 relevant tweets written by 144,741 users between February 1, 2020 and December 31, 2020. The labels from the fact-checking websites were simplified in a binary manner, classifying them as either true or false.

**HoVer** (Jiang et al., 2020): This dataset contains 26,171 claims covering various topics. These claims are derived from question-answer pairs sourced from the HOTPOTQA dataset [4]. Annotators from Appen3 were trained to rewrite these question-answer pairs to a single sentence. To determine the veracity labels, the authors extracted facts from Wikipedia and asked the same annotators to label the claims based on whether they supported them or not.

**IFND** (Sharma & Garg, 2023): The Indian Fake News Dataset (IFND) consists of texts and images collected between 2013 and 2021. These data cover elections, politics, COVID-19, violence, and miscellaneous topics. The veracity of these data is determined based on the media from which they were collected. True claims originate from Tribune, Times Now News, The Statesman, and others, while false claims come from the fact-checked columns of Alt News, Boomlive, and media outlets like The Logical Indian, and News Mobile.

**LIAR** (Wang, 2017): LIAR is a dataset of 12.8K short statements scraped from the API of Politifact, a fact-checking website. These statements were made by politicians and can cover various subjects including the economy, health care, and the job market. All of these political statements were manually labeled by Politifact journalists. The truthfulness ratings consist of six categories: pants-fire, false, barely true, half-true, mostly true, and true.

**LIAR-New** (Pelrine et al., 2023): Liar-New is a dataset containing 1,957 claims scraped from Politifact over a period dating from October 2021 to November 2022. Like Liar, these statements focus on the American political class and encompass various topics including health, the economy, and education. Each claim has also been translated into French by two native speakers. Veracity labels are issued by Politifact's fact-checkers and consist of 6 categories: pants-fire, false, barely true, half-true, mostly true, and true. Unlike Liar, Liar-New features possibility labels (possible, impossible or hard). These labels identify whether claims have enough context to be verified.

**MediaEval** (Boididou et al., 2015): This dataset was made available for the MediaEval 2015 test. It includes tweets and images concerning 11 events, such as Hurricane Sandy, the Boston Marathon bombing, the Sochi Olympics, and the Malaysia Airlines Flight 370. The labeling approach is binary. A tweet is labeled as real if it shares multimedia that accurately represents the referenced event, whereas a tweet is labeled as fake if it shares multimedia content that misrepresents the referenced event.

**MM-COVID** (Li et al., 2020): MM-COVID is a dataset containing claims from 6 languages: English, Spanish, Portuguese, Hindi, French, and Italian. The data and their labels were crawled from fact-cheking agencies and reliable media sources. Each claim was then matched with social media engagements from Twitter users. The labels are binary (real or fake).

**MultiClaim** (Pikuliak et al., 2023) : Multiclaim contains 31,305 claims from social media posts in 39 languages. Each of these claims is associated with an article and a label issued by a fact-checking website. The subjects are diverse, and the database also includes a translation of all claims into English.

**NLP4IF-2021** (Shaar et al., 2021) : NLP4IF-2021 is a database of 3,172 Covid-19 X claims. Three languages are present in NLP4IF-2021: Arabic, Bulgarian and English. The veracity labels are binary (yes or no to the question *To what extent does the tweet appear to contain false information?*) and the dataset also contains other labels covering, for example, its harmfulness, its interest for the general public and its need to be fact-checked by experts.

**PHEME** (Zubiaga et al., 2016): This dataset contains tweets published during five breaking news periods: Charlie Hebdo, Ferguson, Germanwings Crash, Ottawa Shooting, and Sydney Siege. Each tweet is annotated as either a rumor or non-rumor.

**PubHealthTab** (Akhtar et al., 2022): This dataset contains 1,942 real-world claims about public health. These claims are extracted from fact-checking and news review websites. Each claim is

---

[4]https://doi.org/10.18653/v1/D18-1259

associated with a summary of the article, a veracity label, and a justification for that label. The labels are coded into three categories: support, refute or not enough info.

**Rumors** (Tam et al., 2019): Rumors is a dataset containing 1,022 rumors collected between May 1, 2017, and November 1, 2017 from the fact-checking website Snopes. The rumors cover various topics, including politics, fraud, fauxtography, crime, and science. Each claim is also associated with tweets, and the veracity labels are as follows: true, mostly true, mixture, mostly false, false, unproven.

**Snopes Fact-news** (Shekhar, 2020): This dataset is scraped from the fact-checking website Snopes. It contains 4,550 claims, all associated with veracity labels, the origin of the claim, a summary of this origin, and short descriptions of what is true and what is false. The labels are the same as RUMORS, namely true, mostly true, mixture, mostly false, false, unproven.

**TruthSeeker2023** (Dadkhah et al., 2023): TruthSeeker2023 is a dataset of 180,000 coded claims from 2009 to 2022. To collect them, the authors initially crawled 1,400 claims and their ground-truth labels from Politifact. Then, keywords from these claims were used to collect associated tweets, which crowdworkers verified for accuracy. These tweets were labeled based on their corresponding claims from Politifact. TruthSeeker2023 includes two label types: a five-way label (Unknown, Mostly True, True, False, Mostly False) and a three-way label (Unknown, True, False).

**Twitter15** (Ma et al., 2017) : Twitter15 contains 1,490 tweets. To identify fake news, two rumor tracking websites, Snopes and Emergent, were used. Tweets related to these fake news stories were then scraped from Twitter using keywords, and their matches were cross-checked by three researchers. Real news tweets was also collected from Twitter via Twitter's free data stream. It's important to note that this is not the original dataset. The original (Liu et al., 2015) has been re-used by the authors of this new database, who have kept the same name while modifying only the labels. The veracity labels are "true", "false" and "non-rumor". To classify them, Ma et al. (2017) has labeled them according to whether or not the author denies the rumor.

**Twitter16** (Ma et al., 2017): Twitter16 is a dataset containing 818 tweets. Like Twitter15, Twitter16 was reproduced by Ma et al. (2017). For the original dataset (Ma et al., 2016), the authors followed the same data collection procedure as for the original Twitter15, but focused solely on the collection of fake news using Snopes. Ma et al. (2017) have modified the labels, which are true, false, unverified, and non-rumor.

**Verite** (Papadopoulos et al., 2024): VERITE is a dataset containing 1,001 claims and associated images. The data were collected from Snopes and Reuters from January 2001 to January 2023. The topics covered are diverse, including politics, culture, entertainment, business, sports, environment, religion, and more. The labels, derived from fact-checking agencies, are coded into three categories: true, out-of-context, and miscaptioned.

**WICO** (Pogorelov et al., 2021): WICO is a dataset dedicated to COVID-19. It includes 364,325 claims. These claims were collected via the Twitter API from January 17, 2020, to June 30, 2021. Approximately 10,000 tweets are manually annotated with the following labels: 5G conspiracy, other conspiracy, non-conspiracy, and undecidable.

**X-Fact** (Gupta & Srikumar, 2021): X-FACT is a dataset of 31,189 short statements scraped from 85 fact-checking websites. Covering various topics, the data are available in 25 languages, including Arabic, Bengali, French, Hindi, Indonesian, Italian, Spanish, Polish, and Portuguese. The veracity labels indicate a decreasing level of truthfulness: true, mostly true, partly true, mostly false, false, unverifiable, and other.

### A.3    SUPPLEMENT ON KEYWORD ANALYSIS

In Table 7, we show some examples of keywords that could lead to bad classifications. The number under the keywords is the number of times the word appears in claims based on its labels of veracity. We can thus see that there is an absence of true statements referring to Harris or Biden, but many that refer to Trump in the Truthseeker2023 dataset.

Figures 1, 2 and 3 also show the distribution of the 40 most frequent words across the IFND, Truthseeker2023, and Twitter16 datasets. The prevalence of the words in each veracity category was calculated using their relative frequency. A word positioned at x = 1 indicates that it is systematically associated with the veracity category specified by the label.

Table 7: Identification of spuriously predictive keywords.

| | IFND | | | Truthseeker2023 | | | Twitter16 | | |
|---|---|---|---|---|---|---|---|---|---|
| | Fact | Viral | Court | Biden | Harris | Trump | Steve | Potus | Poll |
| True | 17 (0.18%) | 59 (2.06%) | 937 (91.95%) | 44 (0.41%) | 0 (0%) | 6,229 (63.05%) | 0 (0%) | 11 (100%) | 0 (0%) |
| False | 9367 (99.82%) | 2812 (97,94%) | 82 (8.05%) | 10,620 (99.59%) | 2,584 (100%) | 3,651 (36.95%) | 15 (100%) | 0 (0%) | 11 (100%) |
| Other | 0 (0%) | 0 (0%) | 0 (0%) | 0 (0%) | 0 (0%) | 0 (0%) | 0 (0%) | 0 (0%) | 0 (0%) |

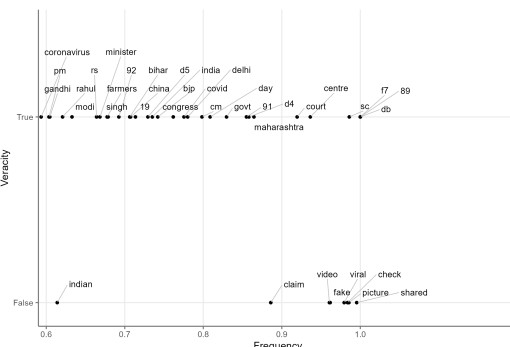

Figure 1: IFND

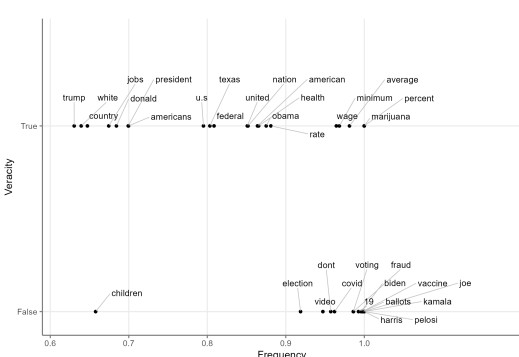

Figure 2: Truthseeker2023

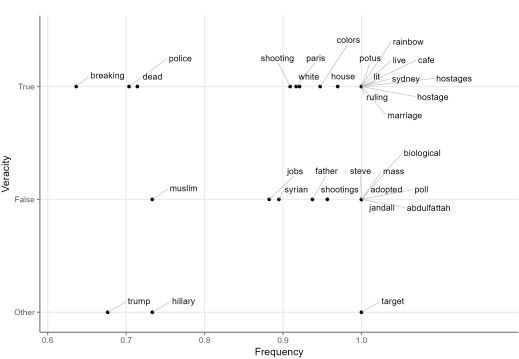

Figure 3: Twitter16

## A.4 SUPPLEMENT ON POLITICAL LEANING

The meaning of ideology and its dimensions varies across countries and regions Caughey et al. (2019); Bauer et al. (2017); Lo et al. (2014). However, most datasets, including the most popular ones on claims, focus on statements made in the U.S. or pertaining to the American context (see Table 1). Given this and the polarization of the American political landscape, we use a partisanship scale that classifies statements as either leaning Republican, leaning Democratic, leaning neither Republican nor Democratic or not political. This allows us to identify the proportion of true and false claims in each dataset that fall within these four categories. More specifically, we use the following prompt on gpt-4-turbo-0409:

```
Statement: {{Statement to analyze}}

Your task is to analyze the partisan lean of different statements. Think step-by-step.
    First, indicate whether the statement is more favorable to Republicans, more
    favorable to Democrats, neither more favorable to Republicans or Democrats or not
    political. For instance, if a statement supports a position commonly associated
    with the Republican Party, it might be more favorable to Republicans. Similarly, if
    a statement highlights a negative characteristic of a Democratic candidate/
    political figure or a positive characteristic of a Republican candidate/political
    figure it might also be more favorable to Republicans.

After providing all of your analysis steps, summarize your analysis and state: "
    Republican Lean; Partisan Lean: (A)" if you think the statement is more favorable
    to Republicans, or "Democratic Lean; Partisan Lean: (B)" if you think the statement
    is more favorable to Democrats, or "Neither Republican or Democratic Lean;
    Partisan Lean: (C)" if you think the political statement is not more favorable to
    Republicans or Democrats, or "Non Political; Partisan Lean: (D)" if you think the
    statement is not political. You should begin your summary with the phrase "Summary
    :"
```

Analyzing the partisan lean of statements is crucial for several reasons. First, providing distributions of the partisan lean enables researchers to select datasets that better align with their specific research goals. For instance, researchers interested in analyzing a balance of Democrat—and Republican—leaning misinformation can choose datasets accordingly. This targeted selection improves the relevance and precision of their studies. Second, if the veracity of a statement correlates with its partisan lean, models may inaccurately assess the statement's veracity based solely on that lean.

Table 8 summarizes our findings. Overall, we observe that a slightly higher proportion of true statements across the datasets are predicted to lean Democratic (12.12%) compared to those leaning Republican (9.5%). In contrast, a greater proportion of false statements are predicted to lean Republican (19.44%) than Democratic (9.36%). However, these proportions vary significantly across datasets. For instance, this partisan bias appears to be more pronounced in some of the most commonly used political datasets, such as LIAR and Twitter 15/16. However, in some datasets—particularly those related to vaccines and COVID-19 (e.g., AntiVax, Check-COVID, MM-COVID)—a larger proportion of statements are classified as neither Democratic nor Republican leaning or as non-political. This is understandable, as anti-vaccine attitudes often emerge from both sides of the political spectrum, especially in the earlier phase of the pandemic (Roberts et al., 2022). Finally, we note that statements that are not explicitly related to politics tend to be more true on average (57.51% vs. 48.28%). This suggests a potential bias on models trained with these data, where political statements may be assumed to be more likely false.

Finally, we note our measurement approach does not account for the context surrounding a claim, it offers a prediction of the partisan lean based solely on the claim itself. Without context, subtleties such as veracity, sarcasm, or nuance might be missed, potentially affecting the predictions. Future research should explore how incorporating additional context might enhance the accuracy and clarity of these predictions. One approach could be to use retrieval-augmented generation by integrating external information from reliable sources for context (Kortukov et al., 2024). Without this added information, models could make incorrect predictions because of the limited context of certain claims. It is thus crucial not only to consider the distribution of the partisan lean when using datasets, but also to recognize that predictive models may lack information on the broader political context when evaluating the veracity of claims.

Table 8: True and False Statements Percentages by Political Leanings

| Dataset | % Rep. Lean (True) | % Dem. Lean (True) | % Neither (True) | % Not Political (True) | % Rep. Lean (False) | % Dem. Lean (False) | % Neither (False) | % Not Political (False) |
|---|---|---|---|---|---|---|---|---|
| IFND | 9.1 | 10.75 | 34.78 | 45.37 | 9.63 | 6.21 | 42.55 | 41.61 |
| AntiVax | 0 | 0 | 0.26 | 99.74 | 0.16 | 0 | 0 | 99.84 |
| Check-COVID | 4.27 | 13.87 | 41.33 | 40.53 | 23.71 | 4.63 | 38.15 | 33.51 |
| Climate-Fever | 21.22 | 41.49 | 20.08 | 17.21 | 67.5 | 8.12 | 11.88 | 12.5 |
| CMU-MisCOV19 | 0 | 0 | 0 | 100 | 0 | 0 | 0.32 | 99.68 |
| CoAID | 3.1 | 8.34 | 44.49 | 44.06 | 52.38 | 4.76 | 28.57 | 14.29 |
| Counter-covid-19-misinformation | 0 | 0 | 0 | 0 | 0 | 0 | 0 | 100 |
| Covid-19-disinformation | 0 | 0 | 0.14 | 99.86 | 0 | 0 | 0 | 100 |
| COVID-19-Rumor | 13.75 | 11.15 | 32.34 | 42.75 | 17.45 | 7.48 | 39.2 | 35.87 |
| COVID-Fact | 3.25 | 6.49 | 29.55 | 60.71 | 5.54 | 5.54 | 34.11 | 54.81 |
| ESOC Covid-19 | 0 | 0 | 0 | 0 | 12.12 | 10.08 | 42.06 | 35.74 |
| FaVIQ | 0.27 | 1.49 | 13.15 | 85.08 | 0.8 | 0.8 | 14.32 | 84.08 |
| FEVER | 1.29 | 1.47 | 14 | 83.24 | 0.94 | 0 | 16.51 | 82.55 |
| FEVEROUS | 0.84 | 1.87 | 11.87 | 85.42 | 2.05 | 1.57 | 14.72 | 81.66 |
| FibVID | 31.67 | 38.33 | 20.83 | 9.17 | 49.93 | 20.53 | 18.01 | 11.52 |
| HoVer | 0.46 | 1.47 | 12.72 | 85.35 | 0.67 | 0.89 | 11.74 | 86.69 |
| LIAR | 37.72 | 31.53 | 24.82 | 5.93 | 51.38 | 25.69 | 18.25 | 4.68 |
| LIAR-New | 44.68 | 34.04 | 19.15 | 2.13 | 53.24 | 16.14 | 19.2 | 11.43 |
| MediaEval | 0.22 | 0 | 0.22 | 99.56 | 0 | 0 | 0.39 | 99.61 |
| MM-COVID | 1.27 | 8.2 | 13.15 | 77.37 | 22.22 | 9.32 | 39.43 | 29.03 |
| PHEME | 17.37 | 22.14 | 32.3 | 28.19 | 10.23 | 18.6 | 37.67 | 33.49 |
| PubHealthTab | 4.9 | 6.15 | 31.47 | 57.48 | 4.44 | 5.92 | 31.95 | 57.69 |
| Rumors | 18.92 | 22.52 | 23.42 | 35.14 | 25.96 | 15.68 | 27 | 31.36 |
| Snopes Fact-news | 21.08 | 22.89 | 28.31 | 27.71 | 28.81 | 15.58 | 30.32 | 25.29 |
| TruthSeeker2023 | 26.37 | 42.97 | 25 | 5.66 | 72.82 | 8.09 | 14.32 | 4.77 |
| Twitter15 | 3.21 | 4.02 | 15.26 | 77.51 | 13.83 | 8.5 | 18.18 | 59.49 |
| Twitter16 | 4.83 | 17.87 | 23.19 | 54.11 | 20.25 | 12.84 | 17.04 | 49.88 |
| Verite | 6.53 | 13.65 | 30.56 | 49.26 | 14.95 | 11.93 | 32.78 | 40.33 |
| WICO | 0 | 0 | 0.31 | 99.69 | 0 | 0 | 0.3 | 99.7 |
| X-Fact | 11.72 | 15.45 | 35.85 | 36.98 | 15.86 | 13.32 | 35.33 | 35.49 |
| Overall | 9.5 | 12.12 | 20.87 | 57.51 | 19.44 | 9.36 | 22.93 | 48.28 |

## A.5 IMPLEMENTATION DETAILS OF GPT-4 WITH WEB SEARCH PREDICTIVE SYSTEM

We implement our web-search predictive system by combining a state-of-the-art "main agent" LLM (OpenAI gpt-4-turbo-0409) with a less powerful but more efficient and cost-effective "search agent" LLM (Cohere command-r). We provide the search agent access to the internet through a Retrieval-Augmented Generation pipeline (RAG, implemented using the Cohere search connector[5].) Specifically, the Cohere search connector applies multiple layers of filtering and reranking to efficiently condense a large number of sources from the web into a succinct response to the query from the main agent. Before any filtering was applied, the total number of tokens retrieved is usually in the range of hundred of thousands of tokens for every single example in the dataset. It would be prohibitively expensive and inefficient if all these sources need to be parsed using the gpt-4-turbo main agent. The summary that the search agent produces, which usually consists of fewer than 200 tokens, is substantially more efficient for the main agent to process while retaining most of the relevant details about the statement.

- Main agent analyzes statement (chain of thought) and proposes queries, if any, to the search agent.
- Search agent:
    - Find relevant documents via open web search. ($\geq$ 100K tokens)
    - Apply re-ranking and filtering. ($\sim$ 50K tokens)
    - Generate condensed response to query. ($\sim$ 200 tokens)
- Main agent analyzes evidences from the search agent. Invoke search agent multiple times as needed.
- Main agent summarizes evidences and draw conclusion.

For further discussion of how this works, please refer to Tian et al. (2024).

## A.6 CONTRADICTION BETWEEN GROUND TRUTH LABEL AND PREDICTIVE SYSTEM

The instances where labelers marked "Predictive system is not wrong," even though the system's output contradicted the ground truth label, can be attributed to differences in timing, interpretation, or problems with the ground truth labels and the claims themselves.

Different timing may lead to contradictions. For instance, in the MM-Covid dataset, there was a claim stating, "Lysol disinfectant label says it was tested against the new coronavirus." The AFP Fact Check labeled this claim as false in September 2020 because, at the time, no Lysol product had been tested against COVID-19. However, a Lysol product was later developed and tested, leading the predictive system to label the claim as true. Similarly, in the LIAR dataset, a claim that "Inflation has gone up every month of the Biden presidency and just hit another 40-year high" was rated as mostly

---

[5]https://docs.cohere.com/docs/overview-rag-connectors

true by PolitiFact in April 2022. However, when the predictive system analyzed the claim using data from January 2024, it labeled it as false, correctly accounting for more recent information.

Another source of contradiction can be the interpretation of the claims. One instance is this claim from the FEVER dataset: "Dakota Fanning is not a model." The ground truth label was false, considering that Dakota Fanning is primarily an actress. However, the predictive system labeled it as true, considering she has engaged in modeling and has appeared in various magazine photoshoots. Here, the system's broader interpretation of what constitutes a "model" led to a contradiction, yet it is not necessarily wrong.

Contradictions also arise due to the specific wording of claims, which is especially prevalent in the MM-Covid dataset. For instance, the ground truth label marked the claim "President Donald Trump's statement that lupus patients are not vulnerable to COVID-19 is not true" as false, focusing solely on Trump's statement. However, the predictive system, which analyzed the entire sentence, classified it as true. The predictive system explained that lupus patients are vulnerable to COVID-19, and thus Donald Trump's statement is indeed not true. Another example is the claim, "These are 6 of the main differences between flu and coronavirus," which had a ground truth label of true based on a headline from the MIT Technology Review. The predictive system, however, labeled it as false, arguing that the differences between the flu and coronavirus cannot be strictly limited to six. The problem is not the labelling of the predictive system, rather the ground truth labels and the claims themselves.

### A.7 SUPPLEMENT ON MANUAL LABELING OF PREDICTION VALIDITY

**LIAR-New** Two authors labeled 100 samples that the GPT-4 (with web search) predictive system got wrong according to standard comparison with the ground truth labels from the professional fact-checkers at PolitiFact (which the dataset is sourced from). The labelers considered the input statement, the reasoning of the predictive system, and the PolitiFact fact-checking article. They each labeled every example, with a 3-way schema: "Predictive system is wrong", "Uncertain / open to interpretation", "Predictive system is not wrong".

This led to 0.36 Cohen Kappa agreement and 60% percentage agreement. The agreement cases within these results indicated a large number of cases where the predictive system was not wrong—38 out of 60 examples where the labels agreed—but to further reinforce the validity of the labeling, the annotators discussed each disagreement and produced a single resolution label. In this final result, of the 100 cases, 30 were "Predictive system is wrong", 15 were "Uncertain / open to interpretation", and the remaining 55 were "Predictive system is not wrong".

The two annotators also manually labeled 100 examples that were originally marked correct. They agreed 76 were not wrong, 2 were uncertain, and 1 was wrong. There were only 5 additional examples that were marked wrong by one but not both annotators.

**FEVER** The same two authors then labeled predictions based on GPT-3.5 (with web search) on the FEVER dataset that were marked incorrect by standard categorical label comparison. Here, there is no fact-checking article to reference, so the authors looked up any necessary information themselves, again seeking to determine if the LLM's explanation was correct. First, they labeled 10 examples together to synchronize the labeling process, then both labeled the same 100 independently. We discard the first 10. On the 100, the labels had 0.51 Cohen Kappa agreement score and 70% agreement. Since the initial agreement was higher, we did not conduct a resolution process on this data. 38 examples were marked "Predictive system is not wrong" by both labelers, and 56 by at least one.

**MM-COVID** Again, the annotators labeled examples that the categorical labels marked incorrect. These were from the GPT-4 (with web search) version of the baseline system. There were only 70 of these total, so the annotators labeled all 70. They had 44% agreement, and agreed that 39 examples were not wrong while agreeing only a mere 3 were wrong. An additional 25 were marked not wrong by one annotator.

Then the annotators labeled 100 examples that were correct according to the categorical labels. They agreed that 89 examples were not wrong, and there were 0 that they agreed were wrong. There were another 5 examples that were marked wrong by one but not both annotators.

### A.8   SUPPLEMENT ON CONTRADICTION EVALUATOR

We implement experiments on the explanations from the GPT-4 with web search predictive system. For comparison with human labels, we use the final version after resolution described above, and drop all "Uncertain / open to interpretation" cases. For all versions of the evaluator, we use GPT-4-Turbo-0409, with temperature 0.0 to reduce variation. There is nonetheless some variation; to further stabilize the estimates, we ran 5 runs and report results using the mean (in the score case) or majority vote (in the binary and trinary cases).

In addition to the score-based version described in the main text, we tested binary and trinary versions of the evaluator. The score-based prompt is:

```
In the following, you will be provided a statement and two assessments of its veracity.
    Your task is to evaluate if the assessments contradict each other. Note that not
    having all of the same evidence or content, or even reaching a different conclusion,
     does not alone constitute a contradiction, especially though not exclusively if
    they are interpreting the statement differently, or considering different time
    periods or other contexts. There's only a contradiction if they actually say
    opposing things that are not up to reasonable interpretation or context differences.

Statement: <statement>

Assessment 1: <article>

Assessment 2: <prediction>

Now that you've ready the statement and assessments, rate how much the assessments
    contradict or not on a scale from 0 (no contradiction) to 10 (complete
    contradiction). However, you must not state your score until you've presented a
    concise analysis. Do not begin your response with a number. First write your
    analysis, then write a vertical bar "|", then finally state your contradiction
    score.
```

Leaving the rest of the prompt unchanged, we adjust the last paragraph as follows to get the binary version:

```
Now that you've ready the statement and assessments, answer if the assessments
    contradict or not. However, you must not state your decision until you've presented
     a concise analysis. Do not begin your response with a label. First write your
    analysis, then write a vertical bar "|", then finally "1: contradiction" or "0: no
    contradiction".
```

And trinary:

```
Now that you've ready the statement and assessments, answer if the assessments
    contradict or not. However, you must not state your decision until you've presented
     a concise analysis. Do not begin your response with a label. First write your
    analysis, then write a vertical bar "|", then finally "1: contradiction" or "0: no
    contradiction", or if you are not sure write "-1: unsure".
```

Binary agrees 68% of the time with the human labels, trinary 67% of the time, and the original score-based approach 68% of the time. Thus, there is little difference in efficacy. We note that the trinary approach, although explicitly given the option to output "unsure", never used it.

## B   NOTE ON COMPUTATIONAL AND OTHER RESOURCES

All political leaning, baselines, and contradiction evaluator experiments were run using the OpenAI API, with the web search version of the latter drawing on Cohere web connector in an agentic setup (Appendix A.5). The total cost was approximately $1,500 USD. Other analyses were run on authors' computers or Google Colab, requiring minimal resources and expenditure.

