# OpenReview forum: "A Guide to Misinformation Detection Datasets"
_ICLR.cc/2025/Conference — ICLR 2025 Conference Withdrawn Submission_

### Official Review · Reviewer_rsLW · 2024-10-18

**Soundness:** 2
**Presentation:** 3
**Contribution:** 2
**Rating:** 3
**Confidence:** 4

**Summary:**

The paper presents a survey of misinformation datasets and an analysis of them. Specifically, the paper presents a brief qualitative description of the 70+ datasets and then goes into more nuanced aspects of the short text datasets, particularly around the correlations of certain words and times and to misinformation. The paper then goes on to show how LLMs can provide meaningful baseline predictions of misinformation in the statements as well as how the reasoning of the LLMs in predicting the labels, and their label predictions can be correct when they disagree with the benchmark label. This result indicates that these datasets can have unrecognized ambiguity in their labels. The paper concludes with a quick investigation of other means of quantifying performance on identifying misinformation by instead using an LLM to rate a claim's entailment with a fact check article as a signal of how misinformed the claim is.

**Strengths:**

The primary strengths of the article are its thoroughness and the originality of its final proposal around metrics for misinformation classification. The article does a great job capturing and documenting a wide swath of misinformation datasets, including multi-modal or image-based ones. It is also through in its evaluation of the short text datasets, looking at known shortcomings such as a keyword or temporal dependence in what is misinformation (e.g., from previous studies like Pennycook et al.).

The primary strength of the article and the part I found most novel and significant was the brief exploration in section 5.2 about why categorical labels can be suboptimal for misinformation – broadly construed – and how maybe evaluating for misinformation as an entailment task with good information (e.g., fact check articles) provides a better signal of misinformation. I think this is significant as it addresses a more fundamental problem of misinformation labeling, namely that misinformation is rarely strongly binary (every good lie has a kernel of truth in it), and that it is intricately tied up in a perception system of other concepts like fake news, narrative belief, frames etc. (see Starbird on this: https://www.cip.uw.edu/2023/12/06/rumors-collective-sensemaking-kate-starbird/). Additionally, I also praise the novelty of approaching the problem like an entailment problem with high-quality facts. Ultimately, I think the type of thinking presented in this paper about misinformation classification being more than a binary task is how real gains will be made in this field.

**Weaknesses:**

The main weakness of the paper is its novelty and its grounding. For novelty, as mentioned throughout the work,  there are some papers already that have done surveys of existing misinformation datasets (e.g., D’Ulizia et al., Sharm et al., etc.). So,  doing another survey, even though it gets more datasets, is not particularly novel. Additionally, the novel elements of the paper, such as finding that binary labels for misinformation are fundamentally flawed, due to the nature of misinformation, is only one small sub-section of the paper and not the focus of the paper.

For grounding, there are a number of assumptions or procedures made throughout that need to be clarified. For one, there is no discussion about the various definitions of misinformation, how they relate to other concepts like disinformation or fake news, and by corollary, how that might result in major differences between datasets. This conflation of misinformation even occurs in the text, on line 214, where the authors briefly switch to discussing disinformation. Additionally, while I generally agree with the argument behind only looking at short-form texts for the experiments and analysis stage, I think the paper could have used something like GPT to produce the fundamental claims from the longer-form texts (i.e., “summarize the claim made in the test relative to ____”), so that those datasets could have been used in the subsequent analyses. Finally, the description of how the labels were simplified across datasets seems like it would oversimplify the labels and create issues with using those simplified labels. As mentioned earlier, there is likely no agreement on what defines misinformation between the datasets and so even though the labels might agree in terms of being true or false for misinformation, it is very possible that those label sets are not labeling the same things.

**Questions:**

-	Could you provide more details on how you normalized the labels between the datasets, specifically accounting for the differences in what is being labeled between the datasets?
-	Did you control for things that were initially misinformation but have become true or vice versa, over time? For example, in the realm of COVID, claiming the virus originated from a lab leak in Wuhan, China was considered by many to be misinformation. However, as time and investigations into the origins have evolved, there is some evidence for the lab leak hypothesis and it is generally not considered misinformation or a conspiracy theory any longer.

---

### Official Review · Reviewer_yDBR · 2024-10-21

**Soundness:** 3
**Presentation:** 3
**Contribution:** 2
**Rating:** 3
**Confidence:** 5

**Summary:**

This paper is a review of existing misinformation datasets. The authors evaluate 75 misinformation datasets, 35 of which contain textual information and are further investigated. In the investigation, the paper first studies data quality in terms of labels, spurious correlations, and political bias. Basically, the analysis results indicate that most existing misinformation datasets yield one or more quality concerns, where some of them should not be used due to severe data collection bias. Moreover, the authors implement an LLM-based detection baseline for benchmarking the misinformation detection problem.

**Strengths:**

I appreciate the hard work in conducting a comprehensive survey on such a huge number of misinformation datasets.

Generally, the insights in the Data Quality Section are useful, especially about the spurious keyword and temporal correlations. Future misinformation research should carefully consider the dataset selection.

**Weaknesses:**

I believe a review paper like this is outside the conference scope. I have conducted a search of the query <dataset> on ICLR published paper websites and most of them are about introducing a new dataset not about a review. Although I understand a comprehensive survey of datasets, along with useful insights, can help the community, I would recommend the authors submit this paper to a workshop or a journal. To convince publication, the authors should explain why the paper is suitable for the conference.

In Section 4.1, the authors list all labeling approaches and discuss their corresponding limitations. However, in the following sections, the authors just utilize the original labels in the evaluation. If some of the labeling methods like the source-based approach are not trustworthy enough [1], the paper still fails to address or even propose a potential method to solve the labeling problem.

In Section 4.4, the authors analyze the partisan and ideological lean of claims via LLMs. I am afraid of the political bias caused by LLMs since existing research [2] finds that LLMs like ChatGPT lean liberal, which may be due to training data bias. In this way, leveraging LLMs to analyze political bias should introduce additional bias.

From Table 6, I do not think prompting LLMs (with or without web search) is a competitive baseline for misinformation detection. Also, since the investigated datasets are usually public, conducting web search provides a cheating opportunity for LLMs to find the answers online. The authors themselves claim that the prediction is only a reasonable lower bound but a terrible upper one. In this way, another misinformation detection method (e.g., [3]) should be considered as a general baseline.

[1] Pennycook, Gordon, and David G. Rand. "The psychology of fake news." Trends in cognitive sciences 25.5 (2021): 388-402.
[2] Rozado, David. "The political biases of chatgpt." Social Sciences 12.3 (2023): 148.
[3] Shu, K., Sliva, A., Wang, S., Tang, J., & Liu, H. (2017). Fake news detection on social media: A data mining perspective. ACM SIGKDD explorations newsletter, 19(1), 22-36.

**Questions:**

My questions generally follow the weaknesses above.

1. What are the unique useful insights of this paper towards misinformation detection literature compared to other surveys?

2. Are there any methods or guidelines to mitigate the labeling problem during data collection?

3. The analysis of political bias is biased itself. Could the authors verify the reliability and robustness of using LLMs to analyze political bias?

4. I would like to see the performance of a more common misinformation (fake news) detection baseline on various datasets.

---

### Official Review · Reviewer_tbGr · 2024-10-31

**Soundness:** 2
**Presentation:** 1
**Contribution:** 2
**Rating:** 3
**Confidence:** 4

**Summary:**

This paper provides an overview and evaluation of misinformation detection datasets, claiming to have curated the largest collection to date. It aims to guide researchers in choosing appropriate datasets for misinformation detection tasks by offering baseline evaluations and insights into dataset quality. The authors argue for the need to improve dataset quality and propose some alternatives for more robust evaluation metrics.

**Strengths:**

1.The authors provide a large dataset collection, which could serve as a useful starting point for researchers in misinformation detection.
2. The paper attempts to standardize a labeling schema, which may facilitate cross-dataset comparisons.

**Weaknesses:**

1.Insufficient Dataset Comparison: While the paper includes a wide array of misinformation datasets, it does not conduct an in-depth comparison across datasets. It lacks a clear answer on which datasets are better suited for specific research questions, leaving readers uncertain on dataset applicability for various use cases.
2.Limited Discussion on Modality and Data Evolution: Misinformation datasets are increasingly multi-modal, with a notable shift towards video and image content, especially with the rise of generative AI. This shift has intensified the challenges associated with misinformation detection. However, the paper largely overlooks this trend and fails to address the particular issues related to handling these data types, which are crucial for future misinformation research.
3.Misalignment with ICLR's Focus: The paper’s emphasis is predominantly on dataset curation and less on methodological or theoretical contributions. As a result, it seems out of place in a conference like ICLR, which traditionally emphasizes advances in machine learning and AI methodologies over dataset-centric surveys.
4.Structure and Flow: Some sections appear repetitive, especially in the discussion of labeling and quality assessment approaches. Consolidating these could enhance readability and conciseness.

**Questions:**

The paper highlights some limitations of LLM-based approaches. However, it does not provide a balanced view of the weaknesses of alternative approaches, such as expert-based annotation. How might we approach balancing the limitations of different methods to create a more robust misinformation detection system?

**Details Of Ethics Concerns:**

No ethics concerns

---

### Official Review · Reviewer_bpye · 2024-11-03

**Soundness:** 2
**Presentation:** 2
**Contribution:** 2
**Rating:** 5
**Confidence:** 4

**Summary:**

This article evaluates the quality of (mis)information datasets in current research, focusing on 35 datasets composed of statements or claims from a larger collection of 75. The authors assess these datasets to identify those suitable for reliable empirical work and highlight issues, such as low-quality labels, spurious correlations, or political bias, which could lead to misleading or non-generalizable results. They also present state-of-the-art benchmarks for these datasets but argue that categorical labels (like "true" or "false") may no longer accurately measure model performance. The article discusses alternative approaches to improve evaluation methods and enhance research reliability in misinformation detection.

**Strengths:**

- Addresses a highly relevant issue.
- Comprehensive dataset analysis: The article provides an extensive evaluation of available misinformation datasets, a valuable resource for researchers in the field.
- Insightful discussion on quality assessment: By critically examining label quality and suggesting alternatives, the authors contribute to more accurate and robust evaluation methods.

**Weaknesses:**

- Narrow focus on claim-based datasets: By concentrating specifically on datasets composed of claims, the article may primarily benefit fact-checking rather than misinformation detection more broadly, limiting its general applicability.
- High level of specificity: The article’s detailed focus on certain datasets makes it less versatile for researchers looking at broader aspects of misinformation detection.
- Insufficient exploration of labeling concepts: It lacks a thorough discussion of foundational concepts for labeling, such as credibility, reliability, and source accuracy, which could enhance understanding of the criteria behind dataset quality.

**Questions:**

This article contributes an organized assessment of misinformation datasets, specifically focusing on collections of claim-based data. Its aim of addressing dataset quality in misinformation research responds to an important need in the field, and the work offers some practical insights into the strengths and weaknesses of current datasets.

However, the article’s scope, while presented as broadly relevant to misinformation detection, is actually quite narrow. By concentrating on claim-based datasets, the study’s utility appears limited primarily to fact-checking applications. This focus does not fully address the larger landscape of misinformation research, such as detecting misleading framing, narrative manipulation, or the subtle ways misinformation can manifest. A more accurate framing of this focus, starting with the article title, could clarify the study's specific relevance and better manage reader expectations.

Additionally, the article falls short in its exploration of critical labeling concepts like source credibility, reliability, and accuracy, which are foundational for dataset quality in this field. Addressing these dimensions in more depth would have strengthened the analysis and provided clearer guidance on what constitutes high-quality data for misinformation detection.

In summary, while the article organizes valuable information on claim-based datasets, its narrow focus and limited discussion of key labeling criteria reduce its broader relevance. Expanding on these areas could make it more meaningful for the diverse needs within misinformation research.

Questions:

- Given your focus on claim-based datasets, how applicable do you see your findings for misinformation detection in contexts beyond fact-checking, such as broader narrative analysis or detecting misleading framing?
- How did you address foundational labeling concepts like credibility, reliability, and source accuracy in your analysis? Do you see these and the concept of objectivity and subjectivity in labeling as important areas for future research, and if so, how might they be integrated into dataset quality assessments?

---

### Author Response · Authors · 2024-12-03

Thank you for the feedback. We will make a number of the improvements suggested to the paper and resubmit. We wish to briefly mention a few of the points:

Regarding depth of analysis of the labeling approaches: we will look for ways to improve this, at least with a deeper analysis, as well as looking for other concrete ways we can evaluate the quality of the labels.

Regarding scope: we believe that considering the many thousands of papers that focus on claims, and that 35 claims datasets alone is more than other reviews include for all datasets, this represents a reasonable scope for a paper. We also believe this is an increasingly important topic, as more attention is devoted to it and new researchers to the area could benefit greatly from a strong reference on the data available and pitfalls to avoid. But we are in the process of adding longer format datasets to our analysis as well, and the next version of the paper should report on those too.

We plan to make improvements to the writing, in several ways. For example, we will work to reduce redundancy and improve general clarity. We will also more precisely define misinformation and how that relates to the data studied. We will also try to provide more direct takeaways on what datasets are strong or questionable, which should reduce the work for the reader to connect the dots and make the paper more informative and practical.

The above list isn't exhaustive, and we will do our best to address all feedback in the next version of the paper. Thanks for your time and consideration!

---

### Note · Authors · 2024-12-03

I have read and agree with the venue's withdrawal policy on behalf of myself and my co-authors.